# Bonding Performance of Surface-Treated Zirconia Cantilevered Resin-Bonded Fixed Dental Prostheses: In Vitro Evaluation and Finite Element Analysis

**DOI:** 10.3390/ma16072646

**Published:** 2023-03-27

**Authors:** Tine Malgaj, Roman Papšík, Anže Abram, Andraž Kocjan, Peter Jevnikar

**Affiliations:** 1Department of Prosthodontics, Faculty of Medicine, University of Ljubljana, Hrvatski trg 6, 1000 Ljubljana, Slovenia; 2Department of Material Science, Montanuniversität Leoben, A-8700 Leoben, Austria; 3Department for Nanostructured Materials, Jožef Stefan Institute, Jamova 39, 1000 Ljubljana, Slovenia

**Keywords:** bonding, dental stress analysis, finite element analysis, materials testing, resin bonded fixed partial denture, resin cements, zirconium dioxide

## Abstract

Debonding of zirconia cantilevered resin-bonded fixed dental prostheses (RBFDPs) remains the main treatment complication, therefore, the present in vitro study aimed to evaluate the effect of different surface pretreatments on the bonding of zirconia RBFDPs. Eighty milled zirconia maxillary central incisors, with complementary zirconia cantilevered RBFDPs, were randomly subjected to four different surface pretreatments (*n* = 20): as-machined (AM); airborne-particle abraded (APA); coated with nanostructured alumina coating (NAC); incisor air-abraded and RBFDP coated (NAC_APA). After bonding, half of each group (*n* = 10) was stored in deionized water (150 days/37 °C), thermocycled (37,500 cycles, 5–55 °C), and cyclically loaded (50 N/1.2 × 10^6^). Load-bearing capacity (LBC) was determined using a quasi-static test. Additionally, finite element analysis (FEA) and fractography were performed. *t*-test and one-way ANOVA were used for statistical-analysis. Before aging, the NAC group provided superior LBC to other groups (*p* < 0.05). After aging, the AM specimens debonded spontaneously, while other groups exhibited comparable LBC (*p* ˃ 0.05). The FEA results correlated with the in vitro experiment and fractography, showing highly stressed areas in the bonding interface, cement layer, and in RBFDP’s retainer wing and connector. The NAC RBFDPs exhibited comparable long-term bonding performance to APA and should be regarded as a zirconia pretreatment alternative to APA.

## 1. Introduction

All-ceramic resin-bonded fixed dental prostheses (RBFDPs) represent an increasingly popular treatment modality for replacing missing anterior teeth. Because of the minimal tooth preparations, high esthetics, and time-efficient treatment, RBFDPs offer several advantages over conventional fixed dental prostheses (FDPs), orthodontic space closure, and implant-supported crowns [1,2,3,4]. However, because of their non-retentive geometry [5,6,7,8,9,10] and the inability to effectively bond to zirconia, the debonding incidence of zirconia RBFDPs remains the main clinical complication, with a five-year incidence of 5.5% [11].

To optimize the bonding to the zirconia retainer wing, a combination of airborne-particle abrasion (APA) and chemical bonding with an adhesive monomer, has been widely advocated [12,13]. However, APA may introduce surface cracks and plastic deformation, leading to zirconia strength degradation and premature failures [14,15,16,17,18], especially in novel translucent zirconia containing 5 mol.% yttria, which hinders the tetragonal-to-monoclinic toughening mechanism [19,20]. Furthermore, zirconia surface changes resulting from APA, lead to lowered translucency [21], which may impair the optical properties of the increasingly used thin translucent zirconia restorations. In light of these concerns, different approaches to pretreating the zirconia bonding surface have been studied [22,23,24,25,26,27,28,29]. Nanostructured alumina coating (NAC), an additive nano-roughening pretreatment, has provided a stronger and durable resin–zirconia bond than APA in vitro [30,31]. It has also been shown that NAC does not impair zirconia’s mechanical nor optical properties [32].

There is a considerable lack of clinical trials evaluating the effect of different surface pretreatments on the zirconia RBFDPs’ performance [7,33,34], and no long-term randomized controlled trials have been reported. In vitro experiments offer a more time and cost-effective evaluation of long-term performance of a prosthetic system. Moreover, many variables can be eliminated, in order to focus on the main clinical problem, such as the zirconia–resin bond in the case of zirconia RBFDPs [11].

Despite the vast amount of in vitro studies evaluating bonding to zirconia [22,23,24,25,26,27,28,29], only a few adopted a clinically relevant experimental geometry, based on non-retentive zirconia RBFDPs as an experimental model. In these studies, different RBFDP framework designs [35,36], materials [37], and preparation geometries [38] were studied, while the authors are unaware of studies evaluating the influence of alternative zirconia pretreatment methods. Moreover, in these tests [35,36], substrate fractures predominated, despite debonding being the main clinical complication, and the experimental models were not verified with additional finite element analysis (FEA) and detailed fractography, as proposed previously [39]. This additional analysis is important to determine whether the debonding of specimens actually occurred due to the failure at the bonding interface, or the crack initiated solely in one of the substrates [39,40].

The present in vitro study aimed to evaluate the effect of different surface pretreatments on the bonding of zirconia RBFDPs, using a tailored experimental model to remove any tooth-related variables. Further, to verify the experimental model and provide a deeper understanding of the failure pattern, the combination of FEA and fractography was conducted. The null hypothesis was that no difference would be detected between the bonding performances of differently pretreated zirconia RBFDPs.

## 2. Materials and Methods

### 2.1. Specimen Preparation

A typodont central maxillary incisor (AG-3Z, Frasaco GmbH, Tettnang, Germany) was prepared for zirconia cantilevered RBFDP, according to recently established guidelines, employing non-retentive geometry (Figure 1a) [41], where a minimal retainer wing thickness of 0.7 mm is recommended. First, cervical and aproximal margins were prepared with a 016 diamond torpedo bur. The height of the distal margin provided 3 mm of connector height. The incisal margin was prepared with a 016 diamond round end taper bur. A small pinhole facilitating stable position of the RBFDP during cementation was prepared centrally using a 012 round diamond bur. A shallow groove was prepared distopalatally using a 016 diamond torpedo bur to provide a sufficient connector dimension. Finally, the palatal surface was smoothed with a flame diamond bur.

The prepared incisor was scanned (Identica T500, MEDIT corp., Seoul, South Korea), and a complementary monolithic RBFDP was designed with a computer-aided design/computer-aided manufacturing (CAD-CAM) software (exocad; exocad GmbH) (Figure 1b). The connector cross-sectional area of the RBFDP was 12.7 mm^2^, and the bonding surface area was calculated to be 32 mm^2^, using a three-dimensional (3D) mesh processing package (MeshLab, ISTI-CNR, Pisa, Italy). Eighty prepared incisors, with complementary eighty zirconia RBFDPs, were milled from pre-sintered 3 mol% yttria-stabilized tetragonal zirconia polycrystal (3Y-TZP) blocks (Ceramill Zolid HT, Amann Girrbach AG, Koblach, Austria), using a CAD-CAM unit (Ceramill Motion 1; Amann Girrbach AG, Koblach, Austria), and thereafter sintered (Ceramill Therm III; Amann Girrbach AG, Koblach, Austria) (Figure 1c).

The specimens were then randomly divided into four groups of twenty, where the bonding surfaces of zirconia abutment teeth and RBFDPs were subjected to different surface pretreatment conditions: left as-machined serving as a control (AM), low-pressure airborne-particle abraded (APA), coated with NAC (NAC), bonding surface of RBFDP’s retainer wing coated with NAC and the bonding surface of the abutment tooth airborne-particle abraded (NAC_APA).

APA was performed with 50 μm alumina particles, at a pressure of 0.1 MPa for 15 s, at a 10 mm distance from the tip of the air abrasion unit. The specimens were then ultrasonically cleaned in 97% ethanol for 3 min. For the coating process, 10 zirconia incisors and 10 zirconia RBFDPs were inserted into a glass beaker of 300 mL aluminate-based precursor solution (VALLBOND, Vall-cer d.o.o., Ljubljana, Slovenia) and boiled for 10 min using a magnetic agitator with a hot plate. The calcination firing was carried out in a laboratory furnace, in atmospheric air at 900 °C and a holding time of 30 min. The coating procedure has been described in detail previously [42].

### 2.2. Surface Roughness Assessment

For each surface treatment, a representative retainer wing was inspected under a scanning electron microscope (SEM) (JSM-7600F, Jeol Ltd., Tokio, Japan). Profile roughness averages (Ra) were measured over evaluation lengths of 3 mm, with a total of five readings per representative retainer wing with a contact profilometer (Talysurf 10, Taylor Hobson, Leicester, United Kingdom).

### 2.3. Bonding

Zirconia RBFDPs were bonded to abutment teeth using a silicone (Putty, GC Europe, Leuven, Belgium) guide, to ensure a stable and accurate position of RBFDP during bonding. The specimens were bonded with a chemically curing resin cement, containing 10-methacryloyloxydecyl dihydrogen phosphate (MDP) adhesive monomer (Panavia 21; Kuraray, Tokyo, Japan). A custom-made alignment apparatus was used to standardize the bonding procedure, providing a repeatable loading axis and loading force of 750 N [43]. The excess cement was removed with a disposable microbrush, and glycerin gel (Oxyguard; Kuraray, Tokyo, Japan) was applied to the margins to block the oxygen inhibition layer. The RBFDPs were loaded for 6 min, allowing the cement to polymerize. Each group was divided into two subgroups of ten specimens each. The first subgroup was stored in deionized water at 37 °C for 24 h, and the second subgroup was subjected to aging.

### 2.4. Aging Protocol

The aging protocol included specimen storage in 37 °C deionized water for 150 days, and subsequent thermal cycling (TC) for 37,500 cycles between 5 °C and 55 °C, with a dwell time of 30 s (Thermocycler THE 1100, SD Mechatronik GmbH, Feldkirchen-Westerham, Germany). After that, specimens were subjected to 1.2 × 10^6^ cycles of cyclic loading, at alternating loads between 5 and 50 N (Instron 8871, Instron Corp, Norwood, MA, USA), at a frequency of 10 Hz.

Before cyclic loading, specimens were inserted into a polyurethane model base, with a longitudinal axis of abutment tooth under 45 degrees. A 0.2 mm thick silicon layer (Fitchecker, GC Europe, Leuven, Belgium) covered the root surface and served as an artificial periodontal membrane, to imitate physiologic tooth mobility [35]. The specimens were loaded under 45 degrees, with load transferred through a steel cylinder ball 2 mm in diameter, positioned palatally, 3 mm below the incisal edge of a pontic (Figure 2). A 1 mm thick layer of tin foil was placed between the loading cylinder and the pontic, to achieve uniform force distribution.

### 2.5. Quasi-Static Loading

The non-aged and aged specimens were statically loaded under 45 degrees (Instron 8871, Instron Corp, Norwood, MA, USA). The load was transferred through a blade positioned perpendicular to the incisal edge of the pontic (Figure 3), at a crosshead speed of 1 mm/min, until debonding. The force during loading was recorded, and the maximum load-bearing capacity (LBC) value was extracted. The complete workflow is summarized in Figure 4.

### 2.6. Microstructural Analyses of Debonded Surfaces

After testing, aged RBFDP retainer wings and abutment teeth bonding surfaces were inspected with a light microscope (LM) (SteREO Discovery.V8, Carl Zeiss AG, Oberkochen, Germany) under ×2.5 magnification. The percentage of each retainer wing’s and abutment tooth’s bonding area covered with cement residue was calculated in a software package (ZEN Digital Imaging for Light Microscopy; Carl Zeiss AG). Scanning electron microscopy (JSM-7600F; Jeol Ltd.), at an accelerating voltage of 5 kV, was used to examine the debonded retainer wings of the representative aged specimen for each experimental group. The representative specimen was determined by the LBC value closest to the mean LBC value for each group.

### 2.7. Finite Element Analysis (FEA) of the Experimental Model

In order to evaluate stresses generated by the debonding force during quasi-static loading, finite element analysis (FEA) was performed. A FEA model, simulating quasi-static loading from the in vitro experiment, was created. Three-dimensional (3D) models of the abutment tooth, RBFPD, and resin cement were created (SpaceClaim 2020 R2, Ansys Inc., Canonsburg, PA, USA), and aligned as three separate bonded bodies with contact. Elastic moduli and Poisson ratios were adopted from previous studies [44]. A slightly thicker cement layer of 0.3 mm was modeled, to achieve proper fitting, without intersections, of the retainer wing and abutment tooth. The FEA model was established in the numerical FEA program (ANSYS 2020 R2, Ansys Inc., Canonsburg, PA, USA). All parts were discretized by a mesh of quadrilateral elements (elements with four triangular sides) with quadratic shape functions, and the mesh contained 6.4 million nodes and 4.3 million elements. The mesh on interfaces was refined, to ensure convergence, and the element size was 500 micrometers in general and 20 micrometers or smaller on interfaces. The interfaces between structures were assumed to be perfectly bonded, with no flaws in the material. All solids were assumed to be homogeneous, isotropic, and linearly elastic throughout the entire deformation. Quasi-static loading of the in vitro experiment was simulated. For computation, the mean LBC for APA, NAC, and NAC_APA groups after aging was chosen, since critical stresses within the restoration and bonding interfaces should be seen at this load [45]. The model of the physical indenter was replaced by a point force of 580 N. The force was applied at the centre of the incisal edge of the pontic and acted at approximately 45 degrees relative to the axis going through the tooth from root to crown. The solution for a static problem with implicit formulation, was obtained using an interactive PCG solver. The maximum first principal stress, for evaluating the tensile stress (MPa), was obtained for the stressed regions of the abutment tooth, RBFDP, and cement layer. In addition, the maximum shear stress (MPa) was obtained at the tooth–cement interface and RBFDP–cement interface.

### 2.8. Statistical Analysis

The sample size was determined by conducting a post hoc power analysis of previous in vitro studies evaluating the differences in bond strength between APA and NAC-prepared zirconia [21,42], using statistical software (G Power 3.1, University Düsseldorf, Germany) [46]. Since the power to detect differences at α = 0.05 level of significance was greater than 0.99, a sample size of 10 specimens was determined. Further, post hoc power analysis of detecting the differences between the non-aged groups was performed, before advancing to the aging of the specimens. The statistical analysis was performed using statistical software (IBM SPSS Statistics, v27.0, IBM Corp, New York, NY, USA). Shapiro–Wilk and Levene tests were performed to assess the assumptions of normality of the data and homogeneity of variances. For each surface pretreatment condition, Student’s *t*-tests of independent samples were performed, to assess the LBC differences between the respective non-aged and aged subgroups. The *p*-values were adjusted using the Bonferroni correction method for the multiple comparisons. The LBC data were then split, according to the aging condition, and a one-way ANOVA and Tukey HSD post hoc test were performed, to assess differences in the profile roughness (Ra), LBC, and the percentage of the bonding area covered with residual cement among the groups (α = 0.05).

## 3. Results

### 3.1. Surface Roughness

SEM micrographs of a representative retainer wing’s bonding surface for each pretreatment are presented in Figure 5. APA significantly increased (*p* < 0.05) the mean profile roughness (Ra) ± standard deviation to 0.25 ± 0.01, while Ra for the AM and NAC bonding surfaces were comparable, measuring 0.22 ± 0.02 and 0.21 ± 0.03, respectively (*p* ˃ 0.05).

### 3.2. Quasi-Static Loading

The calculated power for determining differences among the non-aged groups, at α = 0.05 level of significance, was greater than 0.99 when using 10 specimens, indicating an adequate sample size was used. All specimens failed because of retainer wing debonding, except for one specimen in the aged APA group, where the connector and retainer wing fracture occurred at the load of 524 N. Aging conditions and surface pretreatment had a significant effect on LBC (*p* < 0.05) (Table 1). Before aging, differences in mean LBC between all the experimental groups were detected (*p* < 0.05), where NAC provided the highest (724 ± 58 N) and AM the lowest (361 ± 45 N) mean ±standard deviation LBC. During TC, specimens in the AM group debonded spontaneously. Aging significantly decreased LBC values in the NAC and NAC_APA groups (*p* < 0.05), however, there were no differences between the aged APA, NAC, and NAC_APA groups, with mean LBC ranging between 581 N and 590 N (*p* ˃ 0.05).

### 3.3. Microstructural Analyses of Debonded Surfaces

The AM group exhibited a significantly lower percentage of the bonding area covered with cement residue (*p* < 0.05). In other groups, the sum of the percentages of the bonding area covered with residual cement on the retainer wing and complementary abutment tooth, was approximately 100% (Figure 6). SEM micrographs of the representative retainer wing’s debonded surfaces for each aged group are presented in Figure 6. In the AM group, the retainer wing exhibited smaller areas covered with cement residue (Figure 7a) compared to other groups (Figure 7b,c). In the NAC and NAC_APA groups, a similar failure pattern was revealed, with a thin film of NAC remnants in the areas of adhesive failure (Figure 7c).

### 3.4. FEA Results

The mean LBC for the aged groups was 580 N, and this value was used for the stress computation. The in vitro result, fracture initiation sites, and failure modes were in accordance with the FEA results, verifying the current experimental model. The FEA results of the simulation of quasi-static loading under mean load for aged specimens, are shown in colorimetric stress maps based on MPa (Figure 8). High tensile stresses were measured in the distal area of the cement layer (Figure 8a), well reflected by the area of the cohesive cement fracture observed under SEM (Figure 9a). High tensile stresses were also generated in the RBFDP connector area and the mesial half of the retainer wing (Figure 8b), which is in line with the fracture path of the fractured APA RBFDP observed under SEM (Figure 9b). In the fractured APA specimen, the fractured retainer wing remained bonded to the abutment tooth bonding surface, with a crack propagation starting in the incisal part of the retainer wing and continuing throughout the connector. The highest shear stresses at the pontic–resin interface and tooth–resin interface, were generated in the medial part of the retainer wing (Figure 8c,d), indicating the debonding initiated in the medial part of the bonding interfaces. The stress magnitudes were comparable, whereas slightly larger areas were affected in the RBFDP–resin interface.

## 4. Discussion

Based on the findings of this in vitro study, a superior initial bonding performance of NAC RBFDPs was observed. However, after aging, there was no difference in the bonding performances between the NAC and APA RBFDPs, exhibiting comparable LBC values. Therefore, the null hypothesis was partially accepted. Further, the experimental model was successfully confirmed with FEA.

The bonding capacity of different surface pretreatments was evaluated, using an experimental model based on non-retentive RBFDPs, in order to closely relate to clinical conditions. The study focused on investigating the resin–zirconia interface, since it is considered to be less predictable and has been frequently reported as the predominant failure site of zirconia RBFDPs [32,37,47]. Milled zirconia abutment teeth excluded the enamel–resin interface and standardized the test [48,49], by avoiding natural tooth-related variables such as anatomic variability, different enamel quality, and enamel dehydration. In addition, stable experimental conditions, such as the same bonding surface area and preparation geometry, prevented the variability of stress distributions at the adhesive interface, causing different failure patterns [50].

To in vitro test the bonding capacity of zirconia pretreated with different surface pretreatments, the resin–zirconia interface was maximally stressed, by employing a minimal recommended bonding area [41] and a horizontal loading direction under 45 degrees [35]. During quasi-static loading, the load was transferred on the incisal edge, simulating the worst-case scenario of accidentally biting into a hard bolus [37]. This way, LBC could be compared with maximal incisive forces, which were previously measured in an edge-to-edge position of incisors [51].

Before artificial aging, all pretreatment methods provided clinically adequate bonds, with LBC values exceeding the maximal incisive mastication force (Table 1), which is around 300 N [51]. The NAC group exhibited significantly higher initial LBC values than the APA group (Table 1), exceeding the maximal incisal mastication force by a factor of two. While higher microroughness provided by APA was measured, the NAC’s nanoroughness could not be determined by contact profilometry. However, an approximately 500 nm thick NAC layer provides an increase of the zirconia surface bonding area of 5 to 6 times, facilitating resin cement penetration into nano-scaled inter-lamellar spaces, forming a hybrid layer, which enhanced the resin–zirconia bond [31]. After aging, the AM debonded spontaneously, while LBC in the NAC group significantly decreased, becoming comparable to the clinically accepted APA, which is in line with short-term clinical survival rates of NAC and APA RBFDPs [32].

The negative influence of artificial aging, on the resin bond to NAC-coated zirconia, contrasts with previous findings [30,31,42], which could be ascribed to a highly rigorous aging protocol [52] being adopted in the present study. The generated thermomechanical stresses at the resin–zirconia interface, and a prolonged water exposure facilitating the hydrolysis of the polymer matrix at the resin–zirconia interface [53], might simulate more than five years of RBFDP’s long-term exposure to intraoral conditions [54]. A durable bond in the APA-prepared specimens, corroborates previous findings, where a combination of MDP monomer and APA was shown to facilitate stable resin–zirconia bonds [55,56,57]. While recent findings have reported comparable initial strengths provided by different organophosphate primers, the absence of mechanical roughening led to strength degradation after artificial aging [58].

The predominant debonding failure pattern observed in our study correlated to previous studies, especially when non-retentive preparation geometry was employed. Bishti et al., using the same aging protocol, reported a 75% debonding rate of posterior non-retentive RBFDPs, exhibiting mainly cohesive or mixed failure mode [52]. Sterzenbach et al. loaded zirconia RBFDPs in two stages, increasing the loading force in the second stage, which led to a higher failure rate, with a predominant debonding event [47]. Rosentritt et al. employed similar aging conditions and preparational geometry to our study, and reported an increase in the debonding rate when non-retentive RBFDPs geometry was used [38]. In a study by Brunner et al., zirconia inlay-retained fixed dental prostheses (IRFDP) exhibited the lowest in vitro performance [59], while Gresnigt et al. reported 60% of the specimens debonded during the aging process, despite being properly prepared with APA and MDP monomer [37]. On the contrary, despite the non-retentive preparation geometry used in our study, all of the APA and NAC-treated specimens survived the aging.

Almost complete delamination of the resin cement from both zirconia substrates in the AM group (Figure 6 and Figure 7a), confirmed the necessity of surface roughening and cleaning before priming with adhesive monomer [55,56,57,60]. That there were no differences in the residual cement ratios between the abutment tooth and complementary retainer wing for the NAC and NAC_APA groups (Figure 6), indicates a comparable resin bond to the prepared zirconia substrates after aging. In the NAC and NAC_APA groups, the retainer wing’s areas of adhesive failure were mostly covered with a thin film of NAC remnants (Figure 7c). Consistent with previous studies [31,32], these large areas of NAC residue suggest that the calcination firing protocol that follows NAC synthesis, provided strong bonds between NAC and the zirconia surface.

The in vitro results and failure modes were in accordance with the calculated FEA results, verifying the current experimental model, as recommended previously [39,61]. While critical stresses within restorations did not exceed zirconia fracture toughness, the calculated shear stresses at both interfaces (Figure 8c,d) exceeded previously reported resin–zirconia shear bond strengths to APA and NAC-prepared zirconia [42], explaining the predominant adhesive failure in all the groups. The high debonding rate, in agreement with previous in vitro findings [37,38,47,62], might also be attributed to the experimental model involving stiff zirconia substrates, providing higher stress concentrations in the cement layer and the resin–zirconia interface [63]. In addition, calculated tensile stresses in the lateral area of the retainer wing (Figure 8a) exceeded the reported diametral tensile strength of resin cement [64], facilitating cohesive delamination fracture of the resin cement. This is well reflected by the cement fracture pattern observed in most of the specimens, where a cohesive fracture indeed occurred in the highly stressed area (Figure 9a). Since the stresses in the cement layer increase with the increase of the elastic modulus of the cement, care should be taken in choosing the appropriate cement for bonding zirconia RBFDPs [65].

The aged APA RBFDP, fractured at the connector site involving the retainer wing (Figure 9b). The fracture origin in the retainer wing documented in the present study has rarely been shown in vitro [36]. However, the FEA showed high-stress generation in the medial part of the retainer wing (Figure 8b), in agreement with the present failure pattern. In addition, the minimal thickness of the highly stressed retainer wing, may facilitate such a failure pattern [66]. Although the 3Y-TZP zirconia used in our experiment is less prone to APA’s damaging effect [67], APA might have weakened the mechanical properties of the zirconia retainer wing, causing it to fracture. Furthermore, according to the FEA, tensile stresses in the retainer wing approximated the reported strengths of translucent zirconia with increased yttria concentration of up to 5 mol.%. Since these ceramics are also more susceptible to APA’s damaging effect, APA may be detrimental to RBFDPs fabricated from novel translucent zirconia materials [19,20,68]. In these cases, the use of non-invasive zirconia pretreatment with NAC, providing comparable long-term bonding performance to APA, may be advisable.

The present in vitro study did not include novel translucent zirconia generations as a material for RBFDP fabrication, which may be a limitation. However, the study aimed to evaluate the influence of different surface pretreatments on the bonding properties of zirconia RBFDP, focusing on restoration debonding. Therefore, to minimize the frequency of other failure types, especially restoration fractures, novel translucent zirconia generations with inferior mechanical properties were excluded [19,20]. In addition, it has previously been shown that NAC similarly affects the bonding performance of zirconia containing 3, 4, or 5 mol.% yttria [21].

## 5. Conclusions

Based on the findings of this in vitro study, the following conclusions were drawn:The in vitro result, fracture initiation sites, and failure modes were in accordance with the FEA results, verifying the current experimental model.Both APA and NAC provided an effective long-term bond of resin cement to zirconia RBFDPs, with comparable LBC values (*p* < 0.05) exceeding average and maximum mastication forces.NAC might present a viable non-damaging pretreatment alternative to APA for pretreating monolithic RBFDPs fabricated from more damage-prone translucent zirconia.

## Figures and Tables

**Figure 1 materials-16-02646-f001:**
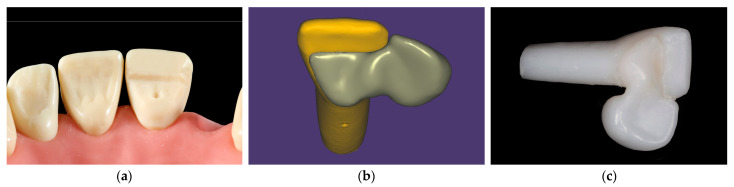
Fabrication of zirconia abutment tooth and zirconia RBFDP. (**a**) Non-retentive preparation of typodont central maxillary incisor for RBFDP; (**b**) scanned abutment tooth and designed complementary zirconia RBFDP; (**c**) milled and sintered zirconia abutment tooth and complementary RBFDP. *RBFPD*, resin-bonded fixed dental prosthesis.

**Figure 2 materials-16-02646-f002:**
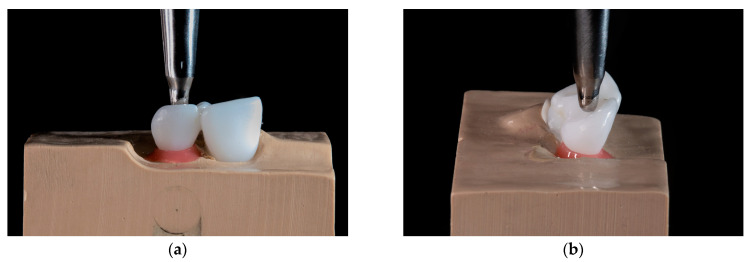
Cyclic loading. The load was transferred through a steel cylinder ball, positioned palatally. (**a**) Frontal view; (**b**) lateral view.

**Figure 3 materials-16-02646-f003:**
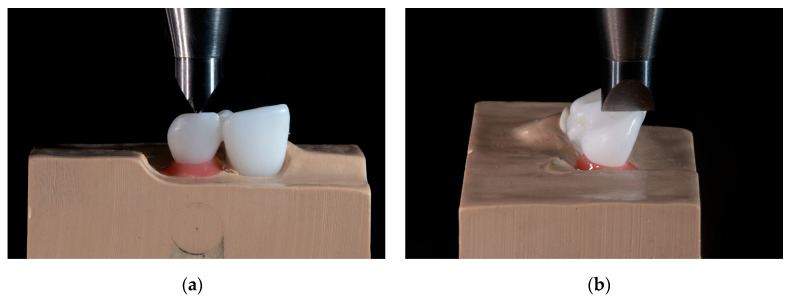
Quasi-static loading. The load was transferred through a blade positioned perpendicular to the incisal edge of the pontic. (**a**) Frontal view; (**b**) lateral view.

**Figure 4 materials-16-02646-f004:**
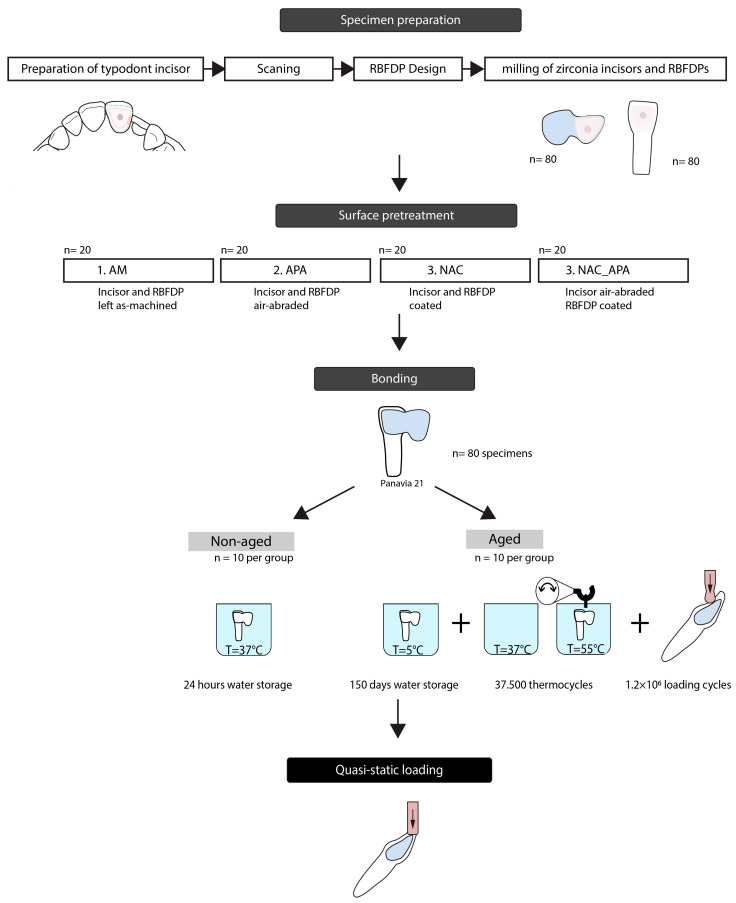
Experimental flowchart.

**Figure 5 materials-16-02646-f005:**
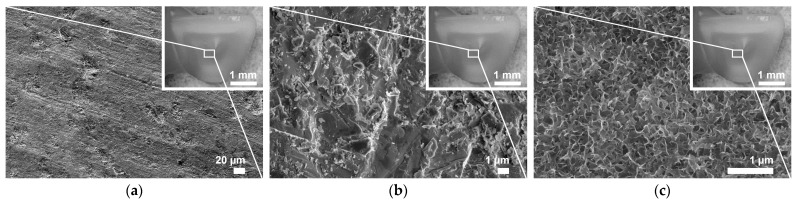
SEM micrographs showing the microstructure of zirconia representative retainer wings’ bonding surface after different pretreatments. (**a**) AM smooth surface with shallow milling traces (original magnification ×250); (**b**) APA surface exhibiting severe surface abrasion, consisting of pits and sharp cuts (original magnification ×5000); (**c**) NAC-modified surface uniformly covered with lamellar-like coating, exhibiting no voids and exposed zirconia grains (original magnification ×20,000). *AM*, as-machined; *APA*, airborne-particle abrasion; *NAC*, nanostructured alumina coating.

**Figure 6 materials-16-02646-f006:**
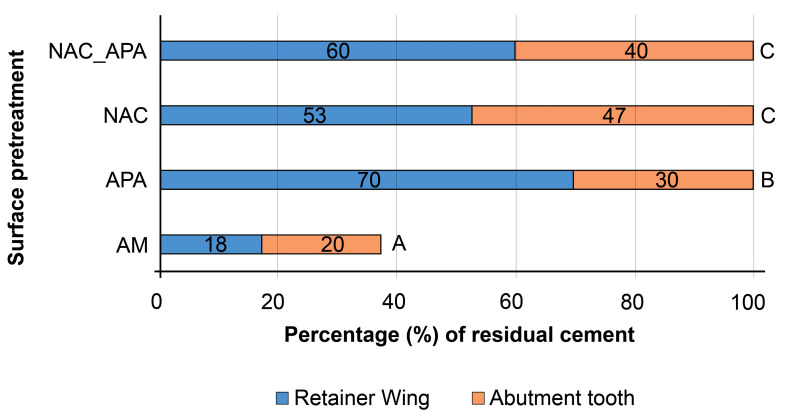
Mean percentages of retainer wing and abutment tooth bonding area covered with cement residue after quasi-static loading of aged groups. The same upper-case letters on the right side of each bar denote no statistical differences between the groups (*p* < 0.05). *AM*, as-machined; *APA*, airborne-particle abrasion; *NAC*, nanostructured alumina coating.

**Figure 7 materials-16-02646-f007:**
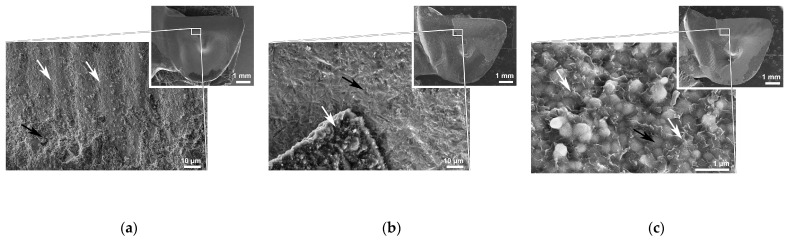
SEM micrographs of representative specimens for each experimental group after aging. (**a**) Low magnification (original magnification ×10) of debonded AM retainer wing (figure inset), showing predominant adhesive failure. Higher magnification (original magnification ×1000) reveals cement residue (black arrow) and shallow milling traces (white arrows). (**b**) Low magnification (original magnification ×10) of debonded APA retainer wing (figure inset) showing areas of complete adhesive failure and areas covered with cement residue. Higher magnification (original magnification ×1000) reveals cement residue (white arrow) and exposed airborne-particle-abraded surface (black arrow). (**c**) Low magnification (original magnification ×10) of debonded NAC retainer wing (figure inset), showing areas of complete adhesive failure and areas covered with cement residue. Higher magnification of adhesive failure area (original magnification ×20,000) reveals NAC residue, with partly exposed zirconia grains covered with a thin film containing remnants of NAC of lamellar-like morphology (white arrow). *AM*, as-machined; *APA*, airborne-particle abrasion; *NAC*, nanostructured alumina coating.

**Figure 8 materials-16-02646-f008:**
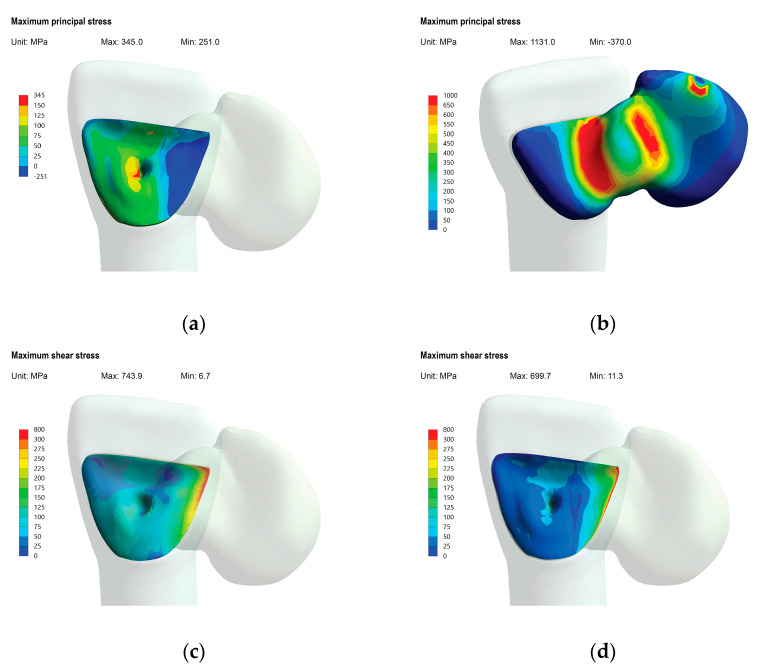
Calculated stresses represented in colorimetric stress maps (MPa). (**a**) Tensile stress in cement layer. (**b**) Tensile stress in RBFDP. (**c**) Shear stress at tooth–resin interface. (**d**) Shear stress at RBFDP–resin interface. *RBFDP*, resin-bonded fixed dental prosthesis; *Max*, maximal stress; *Min*, minimal stress.

**Figure 9 materials-16-02646-f009:**
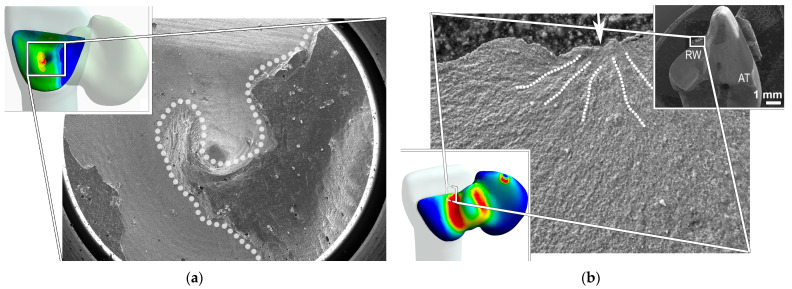
SEM micrographs of fracture patterns relating to calculated stresses. (**a**) Cement fracture pattern (white dotted line) correlating to tensile stress in cement layer (figure inset) (original magnification ×25). (**b**) Retainer wing fracture pattern correlating to tensile stress in RBFDP (left figure inset). At low magnification (right figure inset, original magnification ×8), fractured retainer wing (RW) adhered to the abutment tooth (AT) is observed. A higher magnification (original magnification ×1000) reveals hackles (white dotted lines) pointing to the crack origin in the incisal part of the retainer wing (white arrow). RBFDP, resin-bonded fixed dental prosthesis.

**Table 1 materials-16-02646-t001:** Load-bearing capacity means and standard deviations (SD), in Newtons, of the experimental groups after quasi-static loading. The same upper-case letters in each column denote no statistical differences (*p* < 0.05). * denotes statistically significant difference between LBC values of non-aged and aged subgroups of the same experimental group.

	Non-Aged			Aged			
Group	Mean	SD		Mean	SD		*p* < 0.05
AM	361.4	44.9	A	ds			
APA	564.4	30.6	B	585.2	59.5	A	
NAC	724.1	58.3	C	581.2	60.0	A	*
NAC_APA	654.1	40.7	D	590.3	44.3	A	*

APA, airborne-particle abrasion; NAC, nanostructured alumina coating; SD, standard deviation; ds, debonded spontaneously (no statistical test was performed).

## Data Availability

Not applicable.

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
