# Peer review of "Bonding Performance of Surface-Treated Zirconia Cantilevered Resin-Bonded Fixed Dental Prostheses: In Vitro Evaluation and Finite Element Analysis"

_materials, 2023, doi:10.3390/ma16072646_

Round 1
Reviewer 1 Report
Introduction:
- Please state clearly the aim of the study before the study hypotheses.
Materials and methods:
- ''A typodont central maxillary incisor (AG-3Z; Frasaco GmbH) was prepared for zirconia cantilevered RBFDP according to recently established guidelines employing non- retentive geometry''. Please describe in details the preparation that you performed including the burrs that you used and the thickness you wanted to achieve.
- Did you perform any attempt to standardize the preparation between the 80 teeth? if yes, describe the attempt in the M&M section, otherwise add this point as a limitation of the study.
Discussion and abstract
it is not clear for me why you are talking about initial bonding strength. What test did you perform to evaluate the bond strength prior to aging? Did you divide the samples for the intron machine into 2 subgroups aged and non aged? Please clarify better this part in the m&M section to make it more clear.
- A recent article on the topic can help the reference improvement. Please cite and discuss the following article
Comba, A.; Baldi, A.; Tempesta, R.M.; Carossa, M.; Perrone, L.; Saratti, C.M.; Rocca, G.T.; Femiano, R.; Femiano, F.; Scotti, N. Do Chemical-Based Bonding Techniques Affect the Bond Strength Stability to Cubic Zirconia? Materials2021, 14, 3920. https://doi.org/10.3390/ma14143920
Reviewer 2 Report
A review report of the manuscript titled “Bonding performance of surface-treated zirconia cantilevered resin-bonded fixed dental prostheses: in vitro evaluation and finite element analysis”. Authors of current paper aimed to evaluate the effect of different surface pretreatments on the bonding of zirconia RBFDPs. They concluded that NAC RBFDPs exhibited comparable long-term bonding performance to APA and should be regarded as zirconia pretreatment alternative to APA.
Abstract is informative, but please follow the journal guidelines. If I am not mistaken Abstract should be structures, but without subheadings, so please check and correct accordingly.
Selects the keywords using MESH terms and put them in alphabetical order.
Please provide the sample size calculation.
Please make a flowchart for depicting the whole procedure.
The FEA methodology is not sufficiently described. It should be expanded. There is no information about forces. What type of computational solver (static/dynamic; implicit/explicit)? What about Mesh sensitivity? Many more aspects should be described.
Discussion section should be expanded adding multiple comparisons and discussions with similar studies. I also recommend to add the following papers: https://doi.org/10.3390/ma15124167; https://doi.org/10.3390/molecules28041619
During revision please correct all grammatical and typological errors throughout whole the text
Round 2
Reviewer 1 Report
Dear Authors,
Thank you for addressing my points.
Reviewer 2 Report
Authors addressed most of the concerns and I recommend to add one of the recent papers: https://doi.org/10.3390/molecules28041619